# ROYAL SOCIETY
# OPEN SCIENCE

environmental chemistry

metal-organic framework, adsorption, MCPA, reusability, artificial neural network

**Authors for correspondence:**
Hamza Ahmad Isiyaka
e-mail: hamza_18001996@utp.edu.my
Nonni Soraya Sambudi
e-mail: soraya.sambudi@utp.edu.my

This article has been edited by the Royal Society of Chemistry, including the commissioning, peer review process and editorial aspects up to the point of acceptance.

# Removal of 4-chloro-2-methylphenoxyacetic acid from water by MIL-101(Cr) metal-organic framework: kinetics, isotherms and statistical models

Hamza Ahmad Isiyaka[1], Khairulazhar Jumbri[1], Nonni Soraya Sambudi[2], Zakariyya Uba Zango[1], Bahruddin Saad[1] and Adamu Mustapha[3]

[1]Fundamental and Applied Sciences Department, and [2]Chemical Engineering Department, Universiti Teknologi PETRONAS, Bandar Seri Iskandar 32610, Perak Darul Ridzuan, Malaysia
[3]Department of Geography, Faculty of Earth and Environmental Science, Kano University of Science and Technology, Wudil, 3244 Kano Postal, Nigeria

HAI, 0000-0002-6331-8134; KJ, 0000-0003-3345-6453

Effective removal of 4-chloro-2-methylphenoxyacetic acid (MCPA), an emerging agrochemical contaminant in water with carcinogenic and mutagenic health effects has been reported using hydrothermally synthesized MIL-101(Cr) metal-organic framework (MOF). The properties of the MOF were ascertained using powdered X-ray diffraction (XRD), Fourier transform infrared (FTIR) spectroscopy, thermal gravimetric analysis (TGA), field emission scanning electron microscopy (FESEM) and surface area and porosimetry (SAP). The BET surface area and pore volume of the MOF were 1439 $m^2 g^{-1}$ and 0.77 $cm^3 g^{-1}$, respectively. Artificial neural network (ANN) model was significantly employed for the accurate prediction of the experimental adsorption capacity ($q_e$) values with minimal error. A rapid removal of the pollutant (99%) was recorded within short time (approx. 25 min), and the reusability of the MOF (20 mg) was achieved up to six cycles with over 90% removal efficiency. The kinetics, isotherm and thermodynamics of the process were described by the pseudo-second-order, Freundlich and endothermic adsorption, respectively. The adsorption process is spontaneous based on the negative Gibbs free energy values. The significant correlation between the experimental findings and simulation results suggests the great potential of MIL-101(Cr) for the remediation of MCPA from water matrices.

# ROYAL SOCIETY OF CHEMISTRY

# 1. Introduction

Over the last decades, there has been scientific detection and identification of new emerging pollutants that contaminate surface and groundwater resources [1]. These compounds and their metabolites are found to endanger human health as well as destroy aquatic animals [2]. Evidence from previous studies have shown that agrochemicals such as 4-chloro-2-methylphenoxyacetic acid (MCPA) have immensely contributed to water pollution through agricultural runoff, direct spray and leaching [3,4]. MCPA is among the most extensively used herbicides that have been applied to selectively control weeds in farmlands, roads and gardens [5,6]. It has a high water solubility of 825 mg l$^{-1}$ with a field half-life of 31 days [7]. When excessively applied, their residues may remain in the soil, thereby making it easily available to runoff, and to be leached down to contaminate surface and groundwater [8]. Thus, it is easily transported from a point source to non-point sources over long distance [9]. Some of the physico-chemical properties of the MCPA are highlighted in table 1.

MCPA can easily bioaccumulate and be biomagnified in the tissues of plants and aquatic animals and cause serious health risk to humans [10]. The uncontrolled discharge of this pollutant has significantly contributed to the scarcity of clean water for consumption and other domestic use, particularly in developing countries with high population, agricultural dependent and poor living standard [11]. According to the World Health Organization (WHO), as of 2015, an estimate of 663 million people do not have access to clean water, and by 2050, one in every four persons will consume contaminated water in developing nations [12]. The United States Environmental Protection Agency (USEPA) have listed MCPA as priority pollutant with possible carcinogenic and mutagenic effects [13]. The maximum standard limit for MCPA in drinking water is set by the WHO and USEPA as 0.1 µg l$^{-1}$ [14]. These shortfalls motivate scientists and researchers to search for practical methods for wastewater treatment.

Over the years, various remediation techniques such as coagulation, flocculation, membrane filtration, precipitation, ion exchange, chemical oxidation, biodegradation and adsorption have been applied to remove pollutants from water [15]. Of these methods, adsorption has been singled out to be the most practical and promising technique due to its low cost, simple operations, high selectivity, environmentally benign, convenient recycling and availability of various adsorbents materials [16]. Some adsorbents such as activated [17,18], biochar [19], clay minerals [8] and goethite [20] have been previously used for MCPA adsorption. However, the ideal adsorbent is expected to possess a relatively high surface area, large pore volume, good water and thermal stability under harsh condition [21]. Thus, the quest for finding the best adsorbent materials seems limitless.

Metal-organic frameworks (MOFs) have been substantial applied in different fields due to their peculiar properties for versatile application. They have been considered as one of the highly porous and advanced materials recently discovered [22,23]. MOFs have quickly attracted the attention of researchers for various applications such as $CO_2$ capture, gas storage, sensing, catalysis, biomedical imaging and wastewater remediation [24,25]. They are highly porous, crystalline materials with ultra-high surface area, consisting of metal clusters and multifunctional organic linkers [25].

Among the several MOFs reported, the Materials Institute Lavoisier (MILs) class are exceptionally promising material that have been applied for the remediation of contaminants such as pharmaceuticals, dyes and heavy metals [26,27] in wastewaters. The MIL-101(Cr) is formed from a combination of chromium (III) oxide octahedral trimmers and dicarboxylate linker, resulting in a high class of hybrid supertetrahedron azeotypic mesoporous MOF [28]. The chemical stability of this MOF improves with increasing inertness of the central metal ions and the thermal stability can be explained by the strength of the metal oxygen bond. This is due to the remarkable inertness of Cr(III) that allows the formation of strong Cr-O bond [29]. This MOF is very rigid with large surface area and pore size. MIL-101(Cr) was recently applied by Mirsoleimani-Azizi *et al.* [30] for the removal of pesticide (diazinon) in an aqueous medium. High removal efficiency (92.5%) was achieved. The high adsorption capacity of dyes (377–392 mg g$^{-1}$) was recorded by MIL-101(Cr) with a fast removal efficiency of 99.9% within 30 min [31]. Another study conducted by Gao *et al.* [32] shows rapid removal of pharmaceuticals by MIL-101(Cr) within 60 min. These applications outperformed most of the previous materials applied for the remediation of contaminants in water.

Adsorption process involves nonlinear relationships between several factors that are difficult to model using conventional methods. In previous studies, only one parameter is varied at a time without modelling the simultaneous interaction between the critical adsorption parameters (contact time, initial concentration, dosage, pH and temperature). Limited works have been carried out to

**Table 1.** Some properties of the studied herbicide.

| common name | chemical name | molecular formula | structure | solubility (mg l$^{-1}$) | pKa | log P |
|---|---|---|---|---|---|---|
| MCPA | 4-chloro-2-methylphenoxyacetic acid | $C_9H_9ClO_3$ | | 8.25 | 3.13 | 2.8 |

model and predict the interactive adoption behaviour of MCPA onto MIL-101(Cr). Thus, this study introduces the artificial neural network (ANN) model to evaluate and model the adsorption process and interaction between the adsorption parameters. The ANN is used for the prediction of the experimental findings through learning the pattern of the process. ANN can be trained to develop a non-parametric relationship between multiple input parameters that control the adsorption process [33]. It is a model that mimics the human brain as such does not require prior knowledge of the process controlling the adsorption system. Thus, the aim of this study is to evaluate the removal efficiency of MIL-101(Cr) metal-organic framework for the adsorption of MCPA in an aqueous medium. Batch adsorption experiment has been used to study the effect of the parameters, kinetics and isotherm of the process. ANN was used to model and predict the nonlinear relationship of the adsorption process under the said experimental conditions.

# 2. Material and methods

## 2.1. Materials

Chromium nitrate nonahydrate ($Cr(NO_3)_3 \cdot 9H_2O$, 99%), 1,4-benzene dicarboxylic acid ($H_2BDC$, 99%), N,N-dimethyl formamide (DMF, 99%), methanol (99.5%), ethanol (99.9%) hydrochloric acid and sodium hydroxide were of analytical grade and supplied by Avantis Laboratory (Ipoh Perak, Malaysia) and were used without further purification. MCPA with 98% purity was purchased from Sigma-Aldrich (St Louis, MO, USA).

## 2.2. Synthesis of MIL-101(Cr) adsorbent

MIL-101(Cr) was hydrothermally synthesized according to the previous procedure [34] using $Cr(NO_3)_3 \cdot 9H_2O$ (8 g), $H_2BDC$ (3.32 g) and deionized water (100 ml). The mixture was stirred for 30 min using a magnetic stirrer. HF (10 mmol) was gradually added to the mixture and stirred for 15 min. The mixture was placed in a stainless-steel Teflon-lined autoclave, sealed and inserted into a preheated electric oven at 483 K for 8 h. Next, the autoclave was allowed to cool to room temperature and the product was filtered and recovered. The as-synthesized product was further purified using deionized water, DMF and ethanol to remove possible impurities in the pores. The product yield reached as high as 89%. The purified product was finally dried overnight, cooled to room temperature and stored in a desiccator.

## 2.3. Characterization of MIL-101(Cr) adsorbent

The crystallinity and structural properties of the MIL-101(Cr) were recorded on a Bruker D8 Advance X-ray diffraction (XRD). The thermal stability of the adsorbent was assessed by thermogravimetric analysis (TGA) under $N_2$ atmosphere using Shimadzu TGA-50 Analyser which was heated from 30 to 800°C at a heating rate of 10°C min$^{-1}$. The functional group of the material was determined using Perkin Elmer FTIR Spectrometer which was scanned from 400 to 4000 cm$^{-1}$. Field emission scanning electron microscopy (FESEM) was used to determine the morphology using Zeiss Supra 55 VP

instrument, while the BET surface area and pore size were analysed using $N_2$ adsorption–desorption with Micromeritics ASAP 2020.

## 2.4. Batch adsorption studies

Adsorption experiments were carried out by preparing a stock solution of MCPA (1000 mg l$^{-1}$) by dissolving 100 mg of the analyte in a 1000 ml volumetric flask containing water and was kept in a refrigerator (0°C) prior to use. From the stock, solutions containing different initial concentrations (5–50 mg l$^{-1}$) were studied by dispersing 20 mg of MIL-101(Cr) adsorbent in 100 ml conical flask. The total volume used for each experiment was 50 ml. The flask was then inserted into a thermostatic incubator shaker (incubator ES 20/60, Biosan) and agitated at 150 r.p.m. for 5–60 min. The sample solution (2 ml) was collected and filtered with a nylon syringe membrane (0.45 µm) at every 5 min interval. The absorbance of the MCPA solutions was measured with a UV-Vis spectrophotometer (Shimadzu, Lamda 25). The effects of pH and temperature were studied by adjusting the initial pH from 2 to 12 using either 0.1 M HCl or 0.1 M NaOH, while the temperature was varied from 25 to 50°C. The effect of dosage was also studied by varying the dose from 5 to 50 mg. All the adsorption data were recorded in triplicate from which the average values were calculated.

The quantity of MCPA adsorbed at equilibrium ($q_e$), percentage removal (%R) and quantity adsorbed at a time interval ($q_t$) were calculated using the following equations:

$$q_e = \frac{(C_o - C_e)V}{w} \tag{2.1}$$

$$\%R = \frac{(C_o - C_t)}{C_o} \times 100 \tag{2.2}$$

and

$$q_t = \frac{(C_o - C_t)V}{w}, \tag{2.3}$$

where $C_o$, $C_t$ and $C_e$ are the initial, time and equilibrium concentration of MCPA (mg g−1), $V$ is the volume of the solution (l) and $w$ is the weight of the adsorbent (g).

## 2.5. Artificial neural network model

ANN mimics the behaviour of the human brain in processing information and can learn, predict and correlate the pattern of experimental data when subjected to training [35,36]. The technique provides a platform that can determine the impact of some optimized adsorption parameters in the behaviour of a target output. In this study, the multilayer-perceptron feed-forward-neural network (MLP-FF-ANN) with a back-propagation algorithm and log-sigmoid activation function [37] was used to predict the adsorption capacity of MCPA onto MIL-101(Cr) in correlation with the experimental result. The network structure of the MLP-FF-ANN consists of multiple neurons that are organized in layers. The number of hidden neurons was determined on the basis of trial and error, which forms the training process [38]. The dataset was divided into training (60%), testing (20%) and validation (20%). The network was trained by adjusting the weight to learn the data pattern, and the testing subset was used to evaluate the generalization ability of the network, while the validation datasets were used to estimate the network efficiency. Using these models, criteria such as coefficient of determination ($R^2$), adjusted $R^2$ ($R^2$adj), root mean square error (RMSE) and Akaike information criteria (AIC) are considered as the best fit to judge the performance of our adsorption process by regression analysis. The following equations were used:

$$R^2 = 1 - \frac{\sum (x_i - y_i)^2}{\sum y_i^2 - (\sum y_i^2/n)}, \tag{2.4}$$

$$R^2_{adj} = 1 - (1 - R^2)\left(\frac{n-1}{n-p}\right), \tag{2.5}$$

$$RMSE = \sqrt{\frac{1}{n}\sum_{i=1}^{n}(x_i - y_i)^2} \tag{2.6}$$

and

$$AIC = n\ln\left(\frac{SSE}{n}\right) + 2n_p + \frac{2n_p(n_p + 1)}{n(n_p + 1)}, \tag{2.7}$$

where $x_i$ represents the observed data that was determined experimentally, $y_i$ is the predicted data, $n$ is the number of observation and $p$ denotes the number of parameters.

## 2.6. Adsorption isotherms

Adsorption isotherms provide information on the type of interaction mechanism that exists between MIL-101(Cr) and the studied herbicide. It describes the amount of pollutants adsorbed per unit weight of an adsorbent and the residual pollutant concentration in solution at equilibrium. The Langmuir, Freundlich and Temkin isotherms were used to describe the adsorption of MCPA onto MIL-101(Cr). The models are described by the following equations [39].

Langmuir model

$$\frac{C_e}{q_e} = \frac{1}{K_L q_m} + \frac{C_e}{q_m} \tag{2.8}$$

and

$$R_L = \frac{1}{1 + C_o K_L}, \tag{2.9}$$

where $C_e$ is the concentration at equilibrium (mg g$^{-1}$), $q_e$ is the quantity of MCPA adsorbed at equilibrium (mg g$^{-1}$), $q_m$ and $K_L$ are the constants representing adsorption capacity and adsorption energy, respectively. $R_L$ depicts the favourability of the adsorption process ($R_L > 1$, unfavourable; $0 < R_L < 1$, favourable; $R_L = 1$, linear).

Freundlich model

$$\log (q_e) = \log K_F + \frac{1}{n} \log C_e, \tag{2.10}$$

where $K_F$ is the Freundlich constant of adsorption capacity, $n$ is the adsorption intensity and $C_e$ is the equilibrium concentration of MCPA (mg g$^{-1}$).

Temkin model

$$q_e = B \ln A_T + B \ln C_e, \tag{2.11}$$

where $B$ is the heat of adsorption (J mol$^{-1}$) and $A_T$ is the Temkin equilibrium binding constant corresponding with the maximum binding energy (l g$^{-1}$).

## 2.7. Adsorption kinetics model

The adsorption rate, reaction mechanism and equilibrium time are fundamental in determining the effectiveness and efficiency of the adsorbent material as well as the mass transfer which explains the rate-limiting steps [40]. The pseudo-first-order, pseudo-second-order and intraparticle diffusion model were used to ascertain the best fitting for the experimental data.

Lagergren pseudo-first-order model

$$q_t = q_e(1 - e^{-k_1 t}). \tag{2.12}$$

Pseudo-second-order model

$$q_t = \frac{K_2 q_e^2 t}{1 + K_2 q_e t}. \tag{2.13}$$

Intraparticle diffusion model

$$q_t = K_P t^{0.5} + C, \tag{2.14}$$

where $q_t$ and $q_e$ are the amounts of MCPA adsorbed at certain equilibrium and time $t$ (mg g$^{-1}$), $K_1$ (min$^{-1}$) is the pseudo-first-order rate constant, $K_2$ (g mg$^{-1}$ min$^{-1}$) is the equilibrium rate constant of the pseudo-second-order, and the intraparticle diffusion rate constant is represented as $K_p$ (mg g$^{-1}$ min$^{-1}$).

## 2.8. Thermodynamics studies

The Gibbs free energy change ($\Delta G°$), enthalpy change ($\Delta H°$) and entropy change ($\Delta S°$) were used to determine the feasibility of the adsorption process based on the temperature changes. This helps to

describe whether the adsorption process is spontaneous, exothermic or endothermic. The equations are

$$G° = -RT \ \ln K_C \tag{2.15}$$

and

$$G° = H° - TS°, \tag{2.16}$$

where $\Delta G°$ is the free energy (J K mol$^{-1}$), $T$ (K) and $R$ (J K mol$^{-1}$) are the temperature and universal gas constant for the adsorption, respectively and $K_C$ is the equilibrium constant.

## 2.9. Reusability studies

The reusability of MIL-101(Cr) was studied to assess its potential for regeneration. After the adsorption experiments, the supernatant was decanted, filtered and washed with acetone and distilled water several times. The MOF was vacuum dried for 4 h at 80°C and reused as an adsorbent for the removal of MCPA in water. The process was repeated for six consecutive cycles and the removal of the herbicide was calculated for each cycle.

# 3. Results and discussion

## 3.1. Characterization of MIL-101(Cr)

The diffraction pattern of the obtained MIL-101(Cr) (figure 1a) indicates peaks that are in agreement with those reported in the previous studies [31,41], confirming a well-formed crystalline structure of the MIL-101(Cr). The FTIR spectra of the MOF is presented in figure 1b. The peak at 567 cm$^{-1}$ is attributed to the Cr-O bond which depicts the formation of a well-structured material, and the bands at 746 and 1287 cm$^{-1}$ are assigned to the C-H bond [42]. The peak at 1384 cm$^{-1}$ depicts the symmetric vibration indicating the presence of dicarboxylate group in the MOF [43]. The peak at 1581 cm$^{-1}$ denotes C=C stretching vibration [44], and the strong-broad band at approximately 3433 cm$^{-1}$ shows the presence of O-H group in the material [32]. TGA reveals the thermal stability of the MIL-101(Cr) adsorbent. Three stages of weight loss were observed in figure 1c. The first weight loss is found in the temperature range of 5–200°C. The second weight loss is at approximately 200–308°C, attributed to desorption of adsorbed guest molecules from the pores. The third weight loss (308–600°C) denotes complete decomposition of terephthalic acid in the framework. The MOF was completely decomposed at 800°C, in agreement with an earlier study [41]. The FESEM image of the MIL-101(Cr) (figure 1d) corresponds to an octahedral crystalline structure similar to the previously reported study [42]. The elemental composition at the surface of the MOF as shown in the EDX (figure 1d) contains chromium (38.4%), oxygen (38%) and carbon (23.6%). The surface area and pore size of the MIL-101(Cr) was determined by Brunauer–Emmett–Teller (BET) under N$_2$ adsorption–desorption. The BET surface area of the MOF is approximately 1439 m$^2$ g$^{-1}$ as detailed in table 2.

## 3.2. Artificial neural network prediction model

To develop the best ANN model for accurate prediction requires a careful selection and design of the network architecture, input combinations and model uncertainties [45]. A total of 264 experimental datasets obtained through the Design Expert 11 software were used to train, test and validate the ANN model. Several hidden neurons (table 3) were trained based on a trial and error approach to arrive at the best combination for the prediction of MCPA adsorption capacity, $q_e$ (mg g$^{-1}$). The best ANN architecture comprising 5–8–1 topology (figure 2) was obtained during the training process. The input layer of the selected ANN topology comprises five parameters (contact time, initial concentration, adsorbent dosage, pH and temperature), the hidden layer has eight neurons and the output layer makes up one predicted response (adsorption capacity for MCPA, $q_e$ (mg g$^{-1}$)). The 5–8–1 topology gave the most significant predicted values indicating a better correlation between the experimental and observed datasets with $R^2$ 0.998, 0.999, 0.981 and RMSE 0.088, 0.024, 0.066 for training, testing and validation, respectively (table 3). The scatter plots for the training, testing and validation subsets that show the correlation between the experimental and predicted values are presented in figure 3. The ANN model was also applied to predict the batch adsorption experimental $q_e$ values for each range of experimental conditions in table 4. The result shows a significant

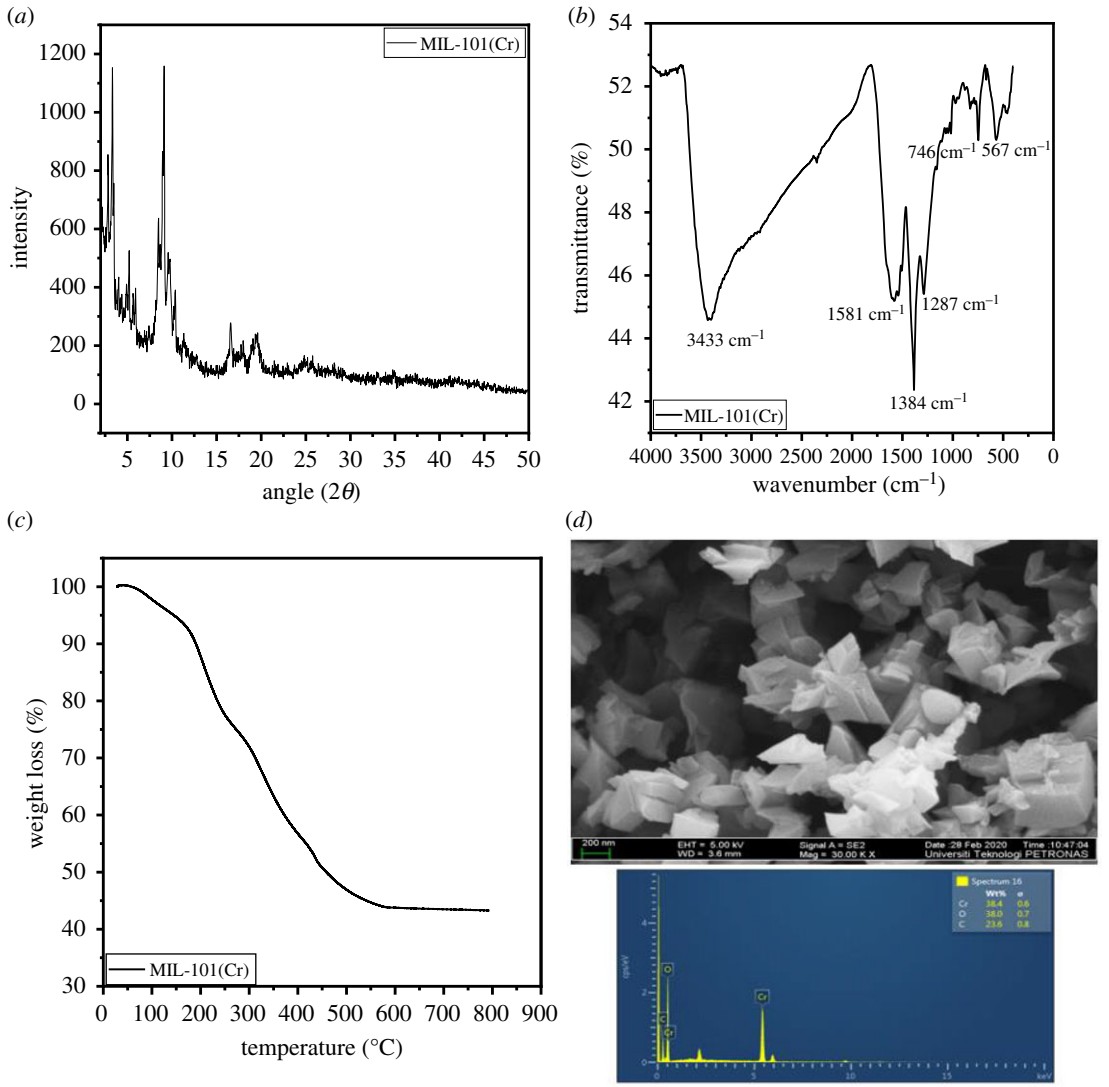

**Figure 1.** (*a*) XRD pattern, (*b*) FTIR spectrum, (*c*) TGA thermogram and (*d*) FESEM-EDX spectrum of MIL-101(Cr).

**Table 2.** BET surface area of MIL-101(Cr) MOF.

| properties | MIL-101 (Cr) |
| --- | --- |
| BET surface area ($m^2\,g^{-1}$) | 1439 |
| Langmuir surface area ($m^2\,g^{-1}$) | 2124 |
| micropore surface area ($m^2\,g^{-1}$) | 182 |
| pore size (nm) | 0.773 |

correlation between the experimental $q_e$ and the predicted $q_e$ values with minimal errors recorded. The predicted $q_e$ is in agreement with the pseudo-second-order kinetics ($q_e$, experimental and $q_e$, calculated), indicating the potential of ANN as a technique for predicting wastewater remediation. This can be attributed to the ability of the ANN to learn the complexity of a dataset when subjected to training and can also model the nonlinear relationship between the actual and predicted variables.

## 3.3. Optimization studies of adsorption parameters

### 3.3.1. Effect of pH in the adsorption process

The initial pH condition of MCPA solution is a key parameter that determines the adsorption process and capacity. In this study, the effect of pH was investigated by varying the pH at the range of 2–12, herbicide

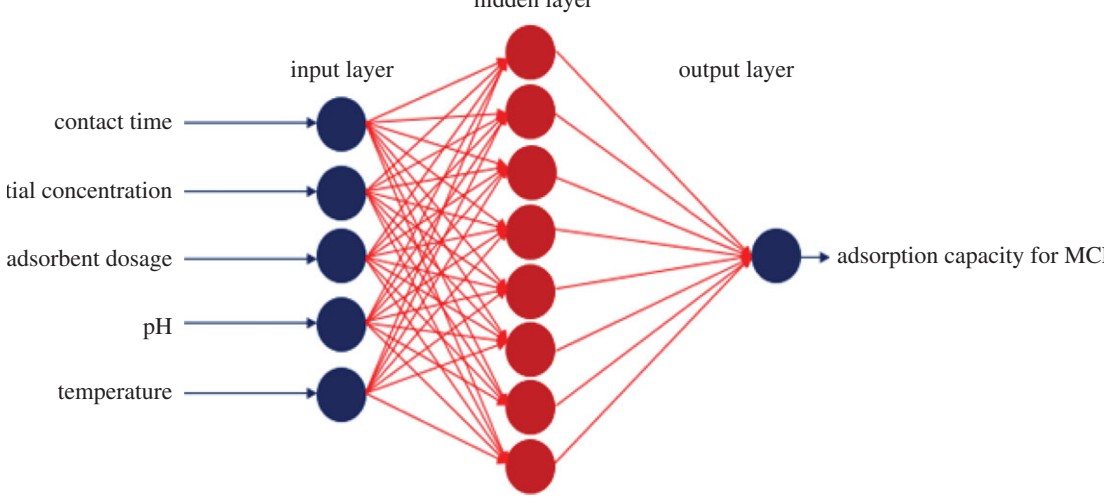

**Figure 2.** Artificial neural network topology.

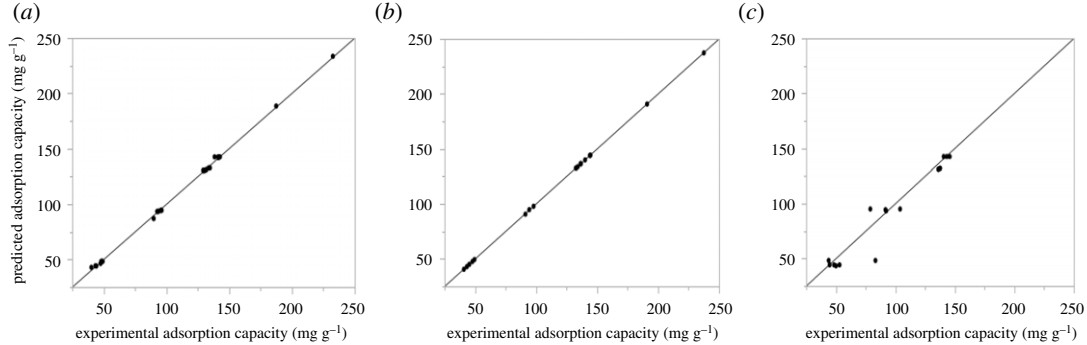

**Figure 3.** Experimental and Predicted ANN scatter plots (*a*) training, (*b*) testing and (*c*) validation.

**Table 3.** Optimum ANN architecture for the prediction of MCPA adsorption capacity.

| no. | neurons | training | | | testing | | | validation | | |
|---|---|---|---|---|---|---|---|---|---|---|
| | | $R^2$ | $R^2$adj | RMSE | $R^2$ | $R^2$adj | RMSE | $R^2$ | $R^2$adj | RMSE |
| 1 | [3] | 0.985 | 0.920 | 1.723 | 0.989 | 0.983 | 1.541 | 0.977 | 0.962 | 1.750 |
| 2 | [4] | 0.996 | 0.953 | 0.686 | 0.997 | 0.994 | 0.612 | 0.978 | 0.971 | 0.608 |
| 3 | [5] | 0.986 | 0.981 | 1.511 | 0.988 | 0.981 | 1.420 | 0.986 | 0.981 | 1.407 |
| 4 | [6] | 0.998 | 0.985 | 1.095 | 0.999 | 0.997 | 1.201 | 0.977 | 0.970 | 1.540 |
| 5 | [7] | 0.997 | 0.991 | 2.188 | 0.998 | 0.995 | 2.321 | 0.967 | 0.918 | 3.053 |
| 6 | [8] | 0.998 | 0.996 | 0.088 | 0.999 | 0.997 | 0.024 | 0.981 | 0.979 | 0.066 |
| 7 | [9] | 0.998 | 0.994 | 1.620 | 0.990 | 0.988 | 1.511 | 0.965 | 0.964 | 1.657 |
| 8 | [10] | 0.996 | 0.987 | 2.797 | 0.989 | 0.981 | 2.833 | 0.982 | 0.981 | 4.729 |
| 9 | [5 5] | 0.988 | 0.913 | 1.668 | 0.932 | 0.931 | 0.321 | 0.941 | 0.930 | 0.944 |
| 10 | [5 7] | 0.934 | 0.922 | 0.392 | 0.912 | 0.908 | 1.443 | 0.910 | 0.906 | 1.321 |
| 11 | [7 6] | 0.911 | 0.902 | 1.866 | 0.905 | 0.910 | 1.832 | 0.909 | 0.899 | 1.612 |

concentration (20 mg l$^{-1}$), dosage (20 mg) and temperature (40°C) as displayed in figure 4. At pH value higher than the pKa of MCPA (3.13), the herbicides solution will be predominantly in the anionic form (negatively charged) due to deprotonation. On the other hand, the surface of MIL-101(Cr) is positively charged at low pH (4–6) and negatively charged when the pH is high (>6). Hence, adsorption at high pH is very low due to the relatively weak interaction between the adsorbent and the herbicide molecule. Thus, a fast and higher removal of 98.62% is observed at

**Table 4.** Comparison between the experimental and predicted MCPA adsorption capacity, $q_e$ (mg g$^{-1}$).

| runs | contact time (min) | initial concentration (mg l$^{-1}$) | adsorbent dosage (mg) | pH | temperature (°C) | experimental, $q_e$ (mg g$^{-1}$) | predicted, $q_e$ (mg g$^{-1}$) | error |
|---|---|---|---|---|---|---|---|---|
| 1 | 15 | 20 | 20 | 4 | 30 | 93.620 | 91.878 | 1.742 |
| 2 | 5 | 10 | 30 | 2 | 35 | 44.388 | 44.208 | 0.180 |
| 3 | 15 | 20 | 40 | 4 | 30 | 94.585 | 94.516 | 0.429 |
| 4 | 15 | 20 | 20 | 8 | 30 | 92.173 | 92.610 | 0.437 |
| 5 | 25 | 40 | 20 | 4 | 30 | 188.577 | 188.445 | 0.132 |
| 6 | 25 | 10 | 10 | 2 | 35 | 48.379 | 49.236 | 0.857 |
| 7 | 15 | 20 | 20 | 4 | 45 | 94.585 | 95.661 | 1.076 |
| 8 | 5 | 30 | 10 | 2 | 35 | 130.441 | 131.085 | 0.644 |
| 9 | 5 | 30 | 10 | 6 | 25 | 132.634 | 133.108 | 0.474 |
| 10 | 5 | 30 | 30 | 2 | 25 | 130.441 | 133.754 | 3.313 |
| 11 | 5 | 10 | 30 | 6 | 35 | 44.388 | 43.845 | 0.543 |
| 12 | 25 | 10 | 10 | 6 | 35 | 48.379 | 48.833 | 0.454 |
| 13 | 25 | 30 | 30 | 6 | 35 | 142.568 | 141.699 | 0.869 |
| 14 | 25 | 30 | 10 | 2 | 35 | 141.910 | 141.187 | 0.723 |
| 15 | 5 | 10 | 10 | 2 | 25 | 44.388 | 44.110 | 0.278 |
| 16 | 5 | 30 | 10 | 6 | 35 | 130.441 | 130.853 | 0.412 |
| 17 | 35 | 20 | 20 | 4 | 30 | 95.243 | 93.853 | 1.390 |
| 18 | 5 | 30 | 30 | 6 | 35 | 130.441 | 131.781 | 1.340 |
| 19 | 5 | 30 | 30 | 6 | 25 | 132.634 | 133.854 | 1.220 |
| 20 | 25 | 10 | 30 | 2 | 25 | 48.379 | 50.190 | 1.811 |
| 21 | 25 | 30 | 10 | 6 | 35 | 142.568 | 141.418 | 1.150 |
| 22 | 15 | 20 | 20 | 10 | 30 | 93.620 | 92.955 | 0.665 |
| 23 | 25 | 10 | 30 | 6 | 25 | 48.379 | 49.095 | 0.716 |
| 24 | 25 | 10 | 30 | 6 | 35 | 48.379 | 48.697 | 0.318 |
| 25 | 45 | 20 | 20 | 4 | 30 | 94.585 | 96.674 | 2.089 |
| 26 | 25 | 20 | 10 | 4 | 40 | 95.243 | 94.028 | 1.557 |
| 27 | 5 | 10 | 10 | 6 | 25 | 44.651 | 43.242 | 1.409 |
| 28 | 5 | 10 | 30 | 6 | 25 | 43.028 | 41.792 | 1.236 |
| 29 | 25 | 30 | 30 | 2 | 35 | 142.568 | 141.741 | 0.827 |
| 30 | 5 | 30 | 10 | 2 | 25 | 131.213 | 133.267 | 2.054 |
| 31 | 25 | 30 | 30 | 2 | 25 | 141.910 | 142.031 | 0.121 |
| 32 | 5 | 10 | 10 | 2 | 35 | 44.388 | 44.869 | 0.481 |
| 33 | 5 | 30 | 30 | 2 | 35 | 132.013 | 131.321 | 0.692 |
| 34 | 25 | 30 | 10 | 6 | 25 | 142.568 | 141.679 | 0.889 |
| 35 | 5 | 10 | 30 | 2 | 25 | 43.537 | 42.995 | 0.542 |
| 36 | 25 | 10 | 10 | 6 | 25 | 48.379 | 48.379 | 0.000 |
| 37 | 15 | 20 | 50 | 4 | 30 | 93.620 | 93.620 | 0.000 |
| 38 | 15 | 20 | 20 | 4 | 30 | 93.620 | 93.620 | 0.000 |
| 39 | 25 | 50 | 10 | 4 | 40 | 233.577 | 233.576 | 0.001 |
| 40 | 5 | 10 | 10 | 6 | 35 | 44.218 | 44.542 | 0.324 |

(*Continued.*)

| runs | contact time (min) | initial concentration (mg l$^{-1}$) | adsorbent dosage (mg) | pH | temperature (°C) | experimental, $q_e$ (mg g$^{-1}$) | predicted, $q_e$ (mg g$^{-1}$) | error |
|---|---|---|---|---|---|---|---|---|
| 41 | 15 | 10 | 10 | 2 | 25 | 46.537 | 46.537 | 0.000 |
| 42 | 25 | 30 | 30 | 6 | 25 | 142.568 | 142.180 | 0.388 |
| 43 | 25 | 30 | 10 | 2 | 25 | 142.568 | 142.568 | 0.000 |
| 44 | 25 | 20 | 10 | 4 | 35 | 48.379 | 48.379 | 0.000 |

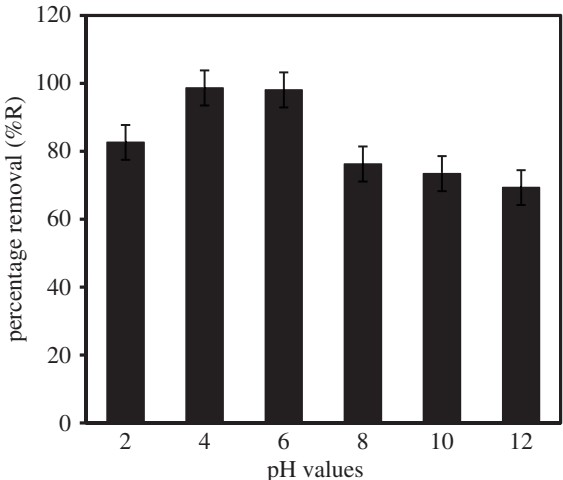

**Figure 4.** Effect of solution pH for MCPA removal (dosage: 20 mg; concentration, 20 mg l$^{-1}$; temperature, 40°C; equilibrium time, 25 min; r.p.m.: 150).

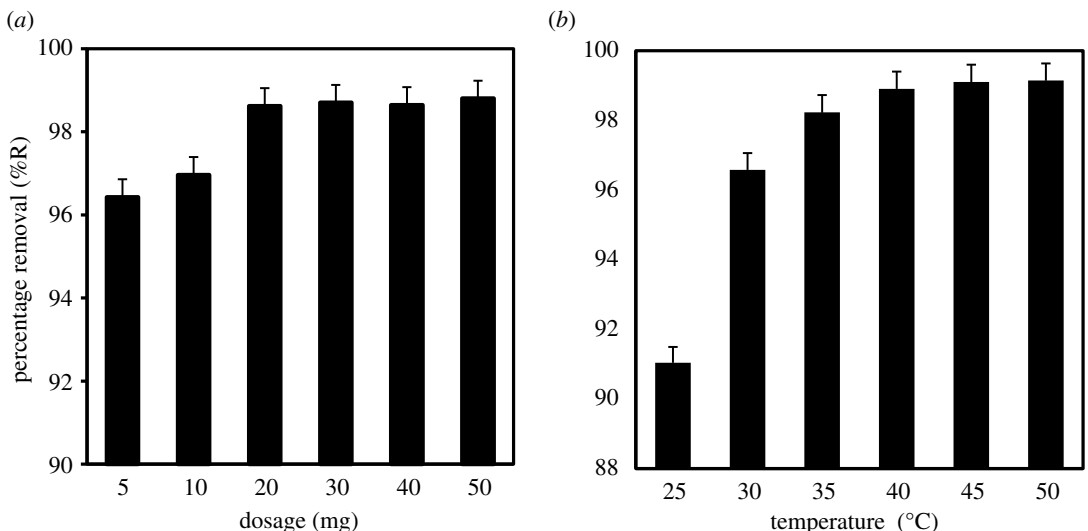

**Figure 5.** (*a*) Effect of adsorbent dose and (*b*) temperature on MCPA removal (concentration, 20 mg l$^{-1}$; pH, 4; equilibrium time, 20 min; r.p.m.: 150).

low pH (4–6) because of the attraction of the negatively charged pollutant in the solution with a positive surface of the MOF adsorbent. Therefore, the mechanism of MCPA adsorption onto MIL-101(Cr) can be attributed to the electrostatic interaction. At neutral and alkaline pH (7–12), the surface of MIL-101(Cr) is negatively charged, leading to electrostatic repulsion with a gradual decrease in the adsorption capacity.

**Table 5.** Thermodynamic parameters for the adsorption of dicamba onto MIL-101(Cr).

| temp (°C) | $\Delta G°$ (kJ mol$^{-1}$) | $\Delta H°$ (kJ mol$^{-1}$) | $\Delta S°$ (kJ mol$^{-1}$ K$^{-1}$) |
|---|---|---|---|
| 25 | −127.856 | 28.334 | 428.952 |
| 30 | −130.007 | | |
| 35 | −132.145 | | |
| 40 | −134.290 | | |
| 45 | −136.753 | | |
| 50 | −138.902 | | |

**Table 6.** Isotherm parameters for adsorption onto MIL-101(Cr).

| isotherm model | parameters | MIL-101(Cr) MCPA |
|---|---|---|
| Langmuir | $q_m$ (mg g$^{-1}$) | 370.37 |
| | $K_L$ (l mg$^{-1}$) | 0.45 |
| | $R_L$ | 0.1 |
| | $R^2$ | 0.92 |
| | $R^2$ adj | 0.899 |
| | RMSE | 0.001 |
| | AIC | −80.57 |
| Freundlich | $K_F$ (mg g$^{-1}$) | 7.524 |
| | $n$ | 1.444 |
| | $R^2$ | 0.999 |
| | $R^2$ adj | 0.997 |
| | RMSE | 0.023 |
| | AIC | −48.017 |
| Temkin | A (mg g$^{-1}$) | 8.253 |
| | bT (kJ mol$^{-1}$) | 62.105 |
| | $R^2$ | 0.914 |
| | $R^2$ adj | 0.893 |
| | RMSE | 26.597 |
| | AIC | 40.937 |

### 3.3.2. Effect of adsorbent dosage

The amount of adsorbent dose that is sufficient for the removal of pollutants in water can be determined by varying the loading. In this study, the dose of MIL-101(Cr) was varied from 5 to 50 mg in 50 ml solution containing 20 mg l$^{-1}$ of MCPA at 40°C. High recovery efficiency (96.4%) was obtained even at the smallest dosage of 5 mg. As the adsorbent dose increases from 10 to 50 mg, the removal efficiency also increased to 98.8% due to the availability of vacant and active adsorption sites. Figure 5a shows that 20 mg is the optimum dose for the removal of MCPA in water. Thus, 20 mg of MIL-101(Cr) adsorbent was adopted for all the subsequent experiments.

### 3.3.3. Effect of temperature and thermodynamics

The temperature at which adsorption takes place plays an important role in the removal process. Thus, the effect of temperature (25–50°C) on the adsorption efficiency was investigated and the results are displayed in figure 5b. The positive correlation between temperature and adsorption efficiency implies that the removal of MCPA increases with an increase in temperature. This is because the increase in

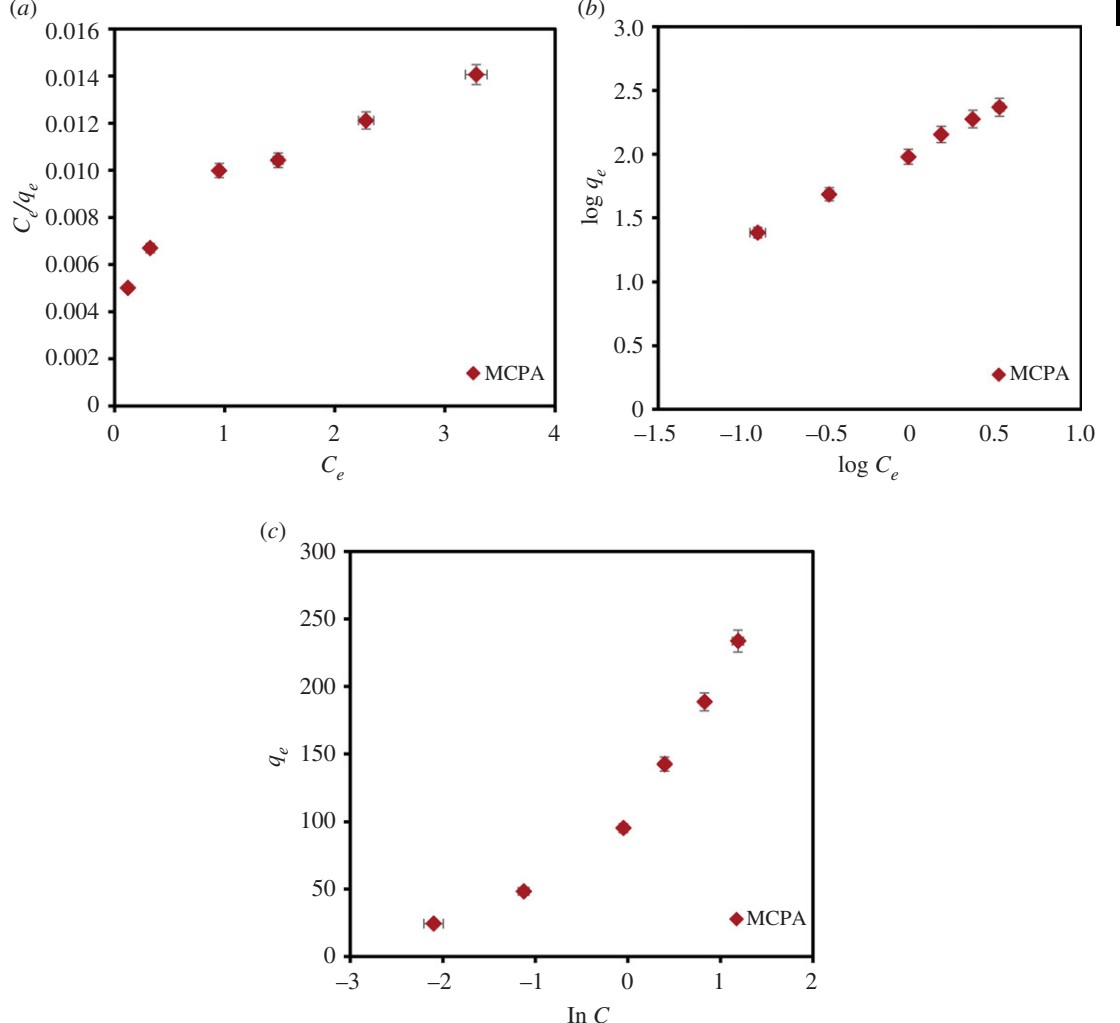

**Figure 6.** (*a*) Langmuir, (*b*) Freundlich and (*c*) Temkin isotherm for the removal of MCPA (dosage: 20 mg; concentration, 20 mg l$^{-1}$; temperature, 40°C; equilibrium time, 20 min; r.p.m.: 150).

the temperature will reduce the viscosity of the solution, which allows easy mobility of the adsorbate molecules [46]. Also, the rise in temperature improves the surface activities and pore capacity of the adsorbent as well as the kinetic energy of the solution. Table 5 shows the thermodynamic parameters. The negative values of the Gibbs free energy (ΔG°) represent a spontaneous adsorption process. The adsorption is endothermic due to the positive enthalpy value.

### 3.3.4. Adsorption isotherm

In this study, the equilibrium data for the removal of MCPA by MIL-101(Cr) were fitted using the Langmuir, Freundlich and Temkin isotherms. The data obtained from the fitted models are presented in table 6 and figure 6*a*–*c*). From the results calculated, the Freundlich isotherm model best fit the adsorption process based on the regression analysis with the highest $R^2$ (0.999), $R^2$adj (0.997); lowest RMSE (0.023); and the least AIC (−48.017) values. The Freundlich model implies an adsorption process on heterogeneous surfaces with binding sites that are not equivalent.

### 3.3.5. Effect of contact time and adsorption kinetics

The impact of contact time in the removal of MCPA by MIL-101(Cr) using different concentrations (5–50 mg l$^{-1}$), contact time (5–60 min), pH (pH 4), adsorbent dose and temperature conditions (pH 3, 20 mg and 40°C) is displayed in figure 7. The results obtained show an excellent and rapid removal of the herbicides within short time (5–10 min) due to the good interaction and porous nature of the MOF

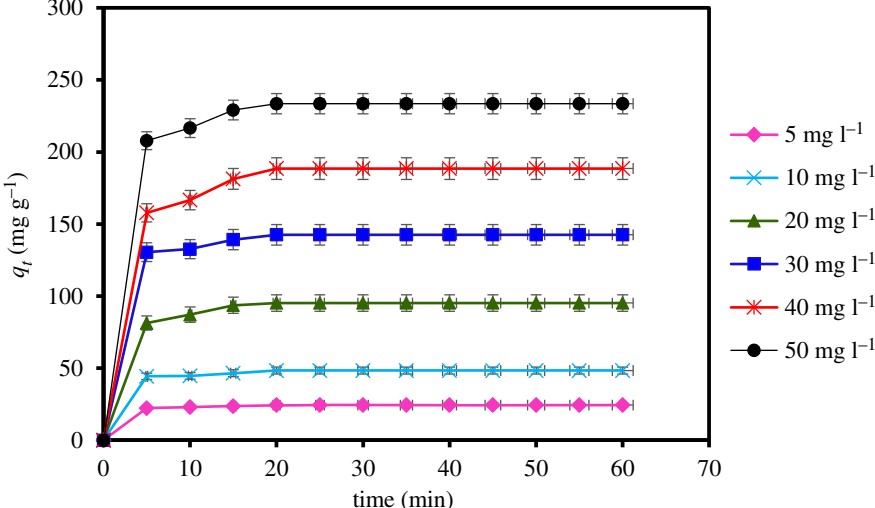

**Figure 7.** Effect of contact time on MCPA removal (dosage: 20 mg; concentration, 5–50 mg l$^{-1}$; temperature, 40°C; equilibrium time, 5–60 min; r.p.m.: 150).

adsorbent. This is corroborated by the high BET surface area of the adsorbent (1439 m$^2$ g$^{-1}$). Hence, equilibrium was attained within 25 min. The contact time was extended until 60 min to ensure the better interactions of the MOF with the analyte after the equilibrium was established.

To better understand the mechanism of adsorption such as a chemical reaction and mass transfer, the kinetics data were fitted using the pseudo-first-order, pseudo-second-order and intraparticle diffusion models. The kinetics results for the models are displayed in table 7 and figure 8a, which indicate that the pseudo-second-order best fit experimental data have the highest coefficient of determination ($R^2 =$ 0.998), $R^2$adj = 0.996, lowest RMSE (0.005) and least AIC (–108.511). This is because the $q_e$ values calculated for the pseudo-second-order kinetic are in agreement with the experimental results. The maximum $q_e$ value for MCPA was determined as 233.576 mg g$^{-1}$ at the equilibrium point, which represents the adsorption capacity of MIL-101(Cr). Thus, the result further explains that the adsorption process is controlled by chemical interaction. The intraparticle diffusion mechanism was also used to describe the interaction and the movement of the molecules inside the particles of the MOF adsorbent. Figure 8b indicates two major stages that represent an external diffusion of the pollutant to the surface of the adsorbent and the transport of the molecules from the surface inside the pore of the MOF. This process describes the rate-limiting step of the adsorption.

### 3.3.6. Reusability studies

The feasibility of MIL-101(Cr) towards the repeated removal of MCPA was studied in view of its regeneration possibility (figure 9). The MOF maintained a steady and high removal efficiency after the third cycle (approx. 98.6%) which indicates better removal capability when compared with the other materials in table 8. A slight decline in the percentage removal (3, 5, 9%) is noticed in the fourth, fifth and sixth cycles, respectively. Nevertheless, the MOF retains approximately 90% removal efficiency after the sixth cycle.

### 3.3.7. Comparison of different adsorbents for the removal of 4-chloro-2-methylphenoxyacetic acid

Different adsorbents that have been applied for the removal of MCPA in water are summarized in table 8. The superiority of the MIL-101(Cr) adsorbent is readily seen, especially in terms of high surface area, adsorption capacity, % removal efficiency (98.6%), fast equilibration time (approx. 25 min) and prospects for regeneration (approx. 90%) after the sixth cycle.

### 3.3.8. Possible adsorption mechanisms

Electrostatic interaction is an important mechanism that determines the adsorptive removal of contaminants in water. The positively charged surface of the MOF can easily interact with the negatively charged adsorbate molecules. The solution pH determines the net surface charge of the adsorbent. The high adsorption capacity of MCPA onto MIL-101(Cr) at low pH is attributed to the

**Table 7.** Adsorption kinetics parameters for the removal of MCPA.

| pseudo-first order | $q_{e,exp}$ (mg g⁻¹) | $q_{e,cal}$ (mg g⁻¹) | $K_1$ (min)⁻¹ | $R^2$ | $R^2$adj | RMSE | AIC |
|---|---|---|---|---|---|---|---|
| mg l⁻¹ | | | | | | | |
| 5 | 24.388 | 9.653 | 0.150 | 0.783 | 0.711 | 0.795 | −0.851 |
| 10 | 48.379 | 23.203 | 0.170 | 0.735 | 0.647 | 0.802 | −0.767 |
| 20 | 95.243 | 66.980 | 0.225 | 0.919 | 0.892 | 0.558 | −4.396 |
| 30 | 142.568 | 84.208 | 0.224 | 0.821 | 0.761 | 0.749 | −1.499 |
| 40 | 188.576 | 169.236 | 0.238 | 0.883 | 0.844 | 0.529 | −4.917 |
| 50 | 233.576 | 170.187 | 0.253 | 0.833 | 0.824 | 0.644 | −2.959 |

| pseudo-second-order MCPA | $q_{e,exp}$ (mg g⁻¹) | $q_e$ cal (g mg⁻¹) | $K_2$ (g mg⁻¹ min⁻¹) | $R^2$ | $R^2$adj | RMSE | AIC |
|---|---|---|---|---|---|---|---|
| 5 | 24.388 | 24.390 | 0.135 | 0.996 | 0.995 | 0.053 | −71.372 |
| 10 | 48.379 | 48.309 | 0.059 | 0.995 | 0.994 | 0.026 | −81.626 |
| 20 | 95.243 | 96.153 | 0.019 | 0.998 | 0.996 | 0.005 | −108.511 |
| 30 | 142.568 | 142.857 | 0.020 | 0.996 | 0.995 | 0.009 | −119.225 |
| 40 | 188.576 | 188.679 | 0.010 | 0.994 | 0.993 | 0.007 | −126.536 |
| 50 | 233.576 | 232.558 | 0.010 | 0994 | 0.993 | 0.006 | −131.463 |

| intrapartide diffusion MCPA | $K_p$ (mg⁻¹ g⁻¹ min$^{1/2}$) | C | $R^2$ | $R^2$adj | RMSE | AIC |
|---|---|---|---|---|---|---|
| 5 | 2.006 | 8.658 | 0.573 | 0.431 | 7.904 | 22.120 |
| 10 | 3.956 | 17.01 | 0.575 | 0.433 | 15.536 | 28.877 |
| 20 | 8.110 | 30.922 | 0.632 | 0.509 | 28.263 | 34.861 |
| 30 | 11.756 | 50.189 | 0.578 | 0.438 | 45.828 | 39.695 |
| 40 | 16.026 | 58.755 | 0.650 | 0.533 | 53.666 | 41.274 |
| 50 | 19.538 | 79.755 | 0.600 | 0.466 | 72.833 | 44.328 |

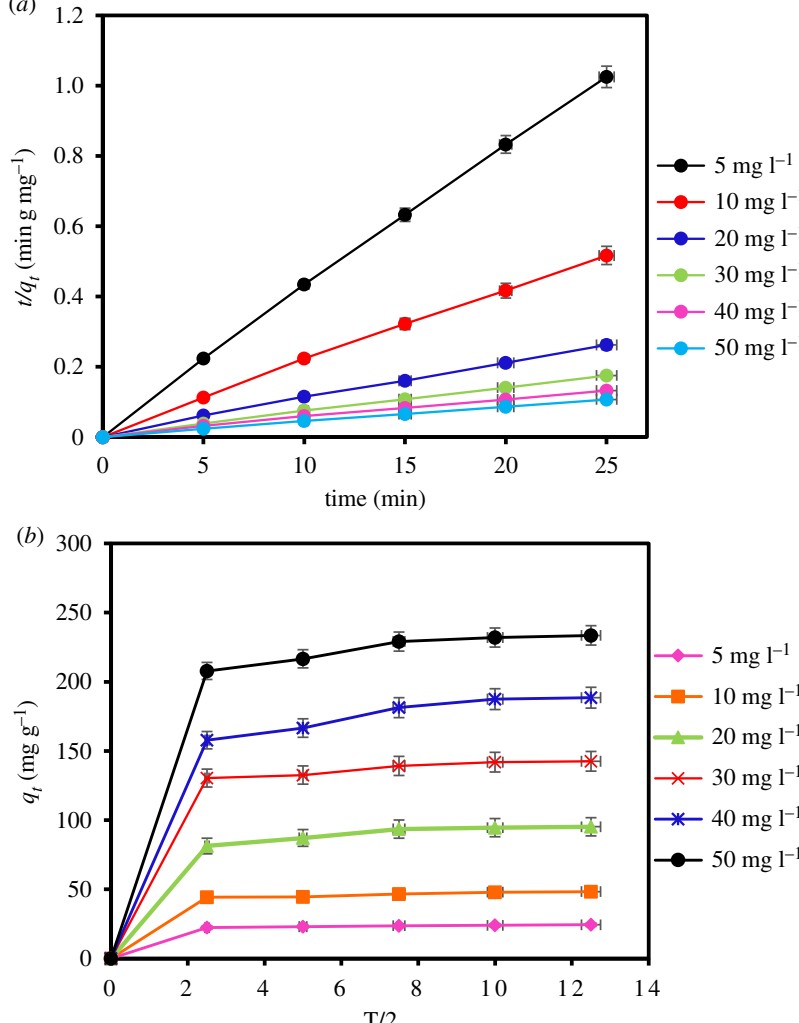

**Figure 8.** (*a*) Pseudo-second-order kinetics and (*b*) intraparticle diffusion model kinetics for MCPA removal (dosage: 20 mg; 40℃; equilibrium time: 25 min; r.p.m.: 150).

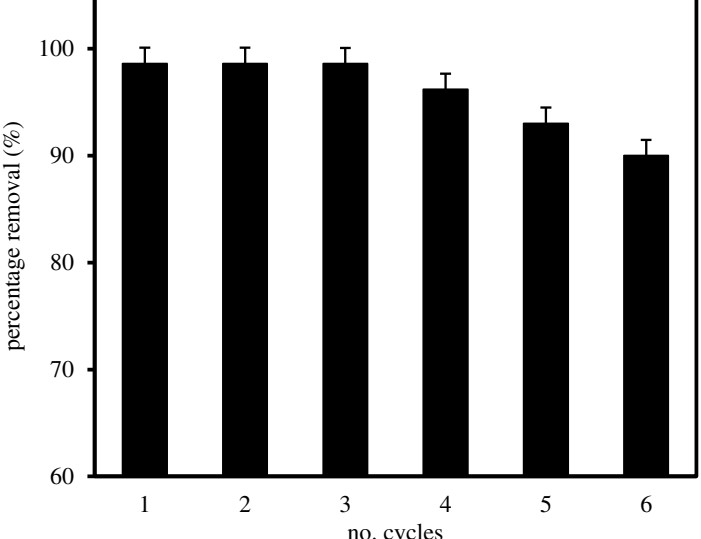

**Figure 9.** Regeneration and reusability potential of MIL-101(Cr) adsorbent.

**Table 8.** Adsorbents reported for the removal of MCPA from water.

| adsorbent | surface area, (m$^2$ g$^{-1}$) | concentrations (mg l$^{-1}$) | (%) R | $Q_e$ (mg g$^{-1}$) | equilibrium time (min) | reuse | ref. |
|---|---|---|---|---|---|---|---|
| activated carbon | 592 | 50 | 70 | 417 | 210 | — | [47] |
| bentonite | 20 | 1 | 80–95 | | 1440 | — | [48] |
| biochar | 1.1 | 100 | 90 | 28 | 360 | — | [49] |
| metal hydroxide | nil | 50 | 84 | 42 | 240 | nil | [50] |
| NH$_2$@COF | 335.7 | | 97.3 | 11 | | | [51] |
| vinylCOF | 345.4 | | 94.3 | 2 | | | |
| MIL-101(Cr) | 1439 | 50 | 98.62 | 234 | 20 | 5 | this work |

electrostatic attraction between the MCPA anions and the positively charged surface of the MOF. As the pH increases, the surface charge gradually decreases causing electrostatic repulsion, thus retarding the adsorption process. The fast and feasible adsorption is also attributed to the chemisorption process which is in agreement with the pseudo-second-order kinetics model. The better fitting with the Freundlich isotherm describes a heterogeneous adsorption surface and an exponential distribution of active sites and energies for better adsorption of MCPA onto the MOF.

## 4. Conclusion

An in-depth assessment of the removal of MCPA using the adsorbent MIL-101(Cr) was conducted by batch experiments. The experimental $q_e$ values for each range of experimental conditions were also predicted by the ANN model with significant correlations and minimal errors. Fast adsorption equilibrium was reached within approximately 25 min using a small dose of the adsorbent material (20 mg). The adsorption process follows the pseudo-second-order kinetics model ($R^2 > 0.998$, RMSE 0.005). The maximum $q_e$ value of the model is 233.576 mg g$^{-1}$. The intraparticle diffusion model indicated a fast phase, signifying an external diffusion of the MCPA molecules from the solution to the surface of the MOF as the rate-determining step, and the slower phase was followed signifying the adsorption of the MCPA molecules to the internal pores of the MOF, until equilibrium is attained. The adsorption process best fit the Freundlich isotherm ($R^2$, 0.999; RMSE, 0.023 and AIC, –48.017). In comparison with other previously reported adsorbent materials, MIL-101(Cr) performed well in the removal of MCPA in terms of fast equilibration, removal efficiency, high adsorption capacity and reusability. It is worth noting that the adsorption process and dataset used for the ANN prediction were based on controlled laboratory conditions.

Ethics. This article does not present research with ethical considerations.

Data accessibility. The datasets supporting this article have been uploaded as part of the electronic supplementary material.

Authors' contributions. B.S. designed the work and proofread the manuscript; H.A.I. and Z.U.Z. carried out the experiment and statistical modelling; K.J., N.S.S. and A.M. co-supervised the project and made a valid contribution to the manuscript.

Competing interests. The authors declare no competing interest.

Funding. This work was funded by the Universiti Teknologi PETRONAS under the YUTP research grant with cost centre 015LCO-211 and UTM CRG Collaborative Research grant no. 015MD0 044.

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
