## [Reviewer comments · Royal Society Open Science]

Review History

RSOS-201553.R0 (Original submission)

Review form: Reviewer 1

Is the manuscript scientifically sound in its present form?

Yes

Are the interpretations and conclusions justified by the results?

Yes

Is the language acceptable?

No

Do you have any ethical concerns with this paper?

No

Have you any concerns about statistical analyses in this paper?

No

Recommendation?

Major revision is needed (please make suggestions in comments)

Comments to the Author(s)

The paper describe the hydrothermal synthesis of MIL-101(Cr) for the removal of 4-chloro-2-methylphenoxyacetic, it's well documented and adsorption mechanism is well explained. Its recommended for publication after Major revision for typo, grammar mistakes and adjusting the following points:

- (WHO) and (USEPA) abbreviations were mentioned in page 3, line 30 however what they stand for is mentioned later in line 34, it should be mentioned earlier with the first use of the abbreviations.
- "4-chloro-2-methylphenoxyacetic" Acid is missing in the first line of the abstract
- It would be nice to add more discussion on MIL-101(Cr) and its application on relevant work and compared it to old used methods in the introduction.
- Page 4, line 42 (environmental benign should be environmentally)
- Page 4, line 46 (hash should be harsh).
- Page 4, line 45 (expected to possessed should be possess)
- Page 3, line 46 punctuation mistake add a comma after Thus.
- Page 5, line 31 (MLP-FF-ANN consist should be consists)
- Page 6, line 2 MIL-10(Cr) should be MIL-101(Cr)
- Page 7, line 8 (formed crystallite should be crystalline)
- Page 7, line 15 (there stages should be three)
- Page 9, line 30 (result show should be result shows)
- Page 12, line 6 (place play should be plays)
- Page 12, line 8 (results is should be results are).
- Page 14, line 50 (stages that represents should be represent)
- Page 16, line 5 (which indicate better should be indicates)
- Page 16, line 7 (the MOF retain should be retains)
- Table 7, MIL-101Cr) should be MIL-101(Cr) line 50 and many more English need to be revised thoroughly.
- Elaborate more in why increasing the temperature increases adsorption efficiency?
- The experimental reaction was conducted between 25-40 °C and TGA showed a weight loss around 5-200 °C, so leaching test to know Cr concentration is recommended since Cr(III) is considered toxic and could encounter oxidizing conditions and the point of this paper is to remove a toxic contaminant form water.

Review form: Reviewer 2

Is the manuscript scientifically sound in its present form?

Yes

Are the interpretations and conclusions justified by the results?

Yes

Is the language acceptable?

Yes

Do you have any ethical concerns with this paper?

No

Have you any concerns about statistical analyses in this paper?

No

Recommendation?

Accept with minor revision (please list in comments)

Comments to the Author(s)

I have read the article entitled "Removal of 4-chloro-2-methylphenoxyacetic acid (MCPA) from water by MIL-101(Cr) metal-organic framework: kinetics, isotherms and statistical models" and my comments as follows:

General comments:

- Generally, the application of MOF as a novel adsorbent for emerging agrochemical contaminant in water is interesting. Nevertheless the author need to highlight the current invention of the recent work done.
- Proposed mechanism need to be discussed in the manuscript.
- It will be best if the author compare the adsorption performance of the synthesized MILs MOF to that of previous work done by others.
- This article suffers minor corrections and the author need to address all my comments before submitting the revised manuscript.

Abstract:

- The author should consider to put numerical value of ΔG_{ads} in the abstract.

Introduction:

- Perhaps some insight the latest inventive step that has been taken in this work. The explanation on the idea of inventive step in this studies could be interesting if it is included in the introduction.
- Novelty need to be highlighted in the last paragraph.

Experimental:

- What is the purity of MCPA used in this study?

Results and discussion:

- Please state the percent yield of the synthesized MILs MOF.
- I guess it would be best if the author compare the adsorption performance of the new MILs MOF from this study with the previous literature (in terms of % MCPA removal).
- Please calculate the ΔG_{ads} value and explain the mode of adsorption. Propose the adsorption mechanism as well.

Conclusion and references:

- Satisfactory

Decision letter (RSOS-201553.R0)

Dear Dr Isiyaka:

Title: Removal of 4-chloro-2-methylphenoxyacetic acid (MCPA) from water by MIL-101(Cr) metal-organic framework: kinetics, isotherms and statistical models
Manuscript ID: RSOS-201553

The editor assigned to your manuscript has now received comments from reviewers. We would like you to revise your paper in accordance with the referee and Subject Editor suggestions which can be found below (not including confidential reports to the Editor). Please note this decision does not guarantee eventual acceptance.

Please submit your revised paper before 19-Nov-2020. Please note that the revision deadline will expire at 00.00am on this date. If we do not hear from you within this time then it will be assumed that the paper has been withdrawn. In exceptional circumstances, extensions may be possible if agreed with the Editorial Office in advance. We do not allow multiple rounds of revision so we urge you to make every effort to fully address all of the comments at this stage. If deemed necessary by the Editors, your manuscript will be sent back to one or more of the original reviewers for assessment. If the original reviewers are not available we may invite new reviewers.

On behalf of the Subject Editor Professor Anthony Stace and the Associate Editor Dr Dattatray Late.

RSC Associate Editor:
Comments to the Author:
Major Revision needed.

RSC Subject Editor:

Comments to the Author:

(There are no comments.)

Reviewers' Comments to Author:

Reviewer: 1

Comments to the Author(s)

The paper describe the hydrothermal synthesis of MIL-101(Cr) for the removal of 4-chloro-2-methylphenoxyacetic, it's well documented and adsorption mechanism is well explained. Its recommended for publication after Major revision for typo, grammar mistakes and adjusting the following points:

- (WHO) and (USEPA) abbreviations were mentioned in page 3, line 30 however what they stand for is mentioned later in line 34, it should be mentioned earlier with the first use of the abbreviations.
- "4-chloro-2-methylphenoxyacetic" Acid is missing in the first line of the abstract
- It would be nice to add more discussion on MIL-101(Cr) and its application on relevant work and compared it to old used methods in the introduction.
- Page 4, line 42 (environmental benign should be environmentally)
- Page 4, line 46 (hash should be harsh).
- Page 4, line 45 (expected to possessed should be possess)
- Page 3, line 46 punctuation mistake add a comma after Thus.
- Page 5, line 31 (MLP-FF-ANN consist should be consists)
- Page 6, line 2 MIL-10(Cr) should be MIL-101(Cr)
- Page 7, line 8 (formed crystallite should be crystalline)
- Page 7, line 15 (there stages should be three)
- Page 9, line 30 (result show should be result shows)
- Page 12, line 6 (place play should be plays)
- Page 12, line 8 (results is should be results are).
- Page 14, line 50 (stages that represents should be represent)
- Page 16, line 5 (which indicate better should be indicates)
- Page 16, line 7 (the MOF retain should be retains)
- Table 7, MIL-101Cr) should be MIL-101(Cr) line 50 and many more English need to be revised thoroughly.
- Elaborate more in why increasing the temperature increases adsorption efficiency?
- The experimental reaction was conducted between 25-40 °C and TGA showed a weight loss around 5-200 °C, so leaching test to know Cr concentration is recommended since Cr(III) is considered toxic and could encounter oxidizing conditions and the point of this paper is to remove a toxic contaminant form water.

Reviewer: 2

Comments to the Author(s)

I have read the article entitled "Removal of 4-chloro-2-methylphenoxyacetic acid (MCPA) from water by MIL-101(Cr) metal-organic framework: kinetics, isotherms and statistical models" and my comments as follows:

General comments:

- Generally, the application of MOF as a novel adsorbent for emerging agrochemical contaminant in water is interesting. Nevertheless the author need to highlight the current invention of the recent work done.

- Proposed mechanism need to be discussed in the manuscript.
- It will be best if the author compare the adsorption performance of the synthesized MILs MOF to that of previous work done by others.
- This article suffers minor corrections and the author need to address all my comments before submitting the revised manuscript.

Abstract:

- The author should consider to put numerical value of ΔG_{ads} in the abstract.

Introduction:

- Perhaps some insight the latest inventive step that has been taken in this work. The explanation on the idea of inventive step in this studies could be interesting if it is included in the introduction.
- Novelty need to be highlighted in the last paragraph.

Experimental:

- What is the purity of MCPA used in this study?

Results and discussion:

- Please state the percent yield of the synthesized MILs MOF.
- I guess it would be best if the author compare the adsorption performance of the new MILs MOF from this study with the previous literature (in terms of % MCPA removal).
- Please calculate the ΔG_{ads} value and explain the mode of adsorption. Propose the adsorption mechanism as well.

Conclusion and references:

- Satisfactory

Author's Response to Decision Letter for (RSOS-201553.R0)

See Appendix A.

Decision letter (RSOS-201553.R1)

Dear Dr Isiyaka:

Title: Removal of 4-chloro-2-methylphenoxyacetic acid (MCPA) from water by MIL-101(Cr) metal-organic framework: kinetics, isotherms and statistical models
Manuscript ID: RSOS-201553.R1

It is a pleasure to accept your manuscript in its current form for publication in Royal Society Open Science. The chemistry content of Royal Society Open Science is published in collaboration with the Royal Society of Chemistry.

On behalf of the Subject Editor Professor Anthony Stace and the Associate Editor Dr Dattatray Late.

RSC Associate Editor
Comments to the Author:
Authors have considered each of the referee's comments carefully. Now manuscript is suitable for publication.

Reviewer(s)' Comments to Author:

Appendix A

UNIVERSITI
TEKNOLOGI
PETRONAS

The Chief Editor,

Royal Society Open Science.

Dear Sir,

RE: Removal of 4-chloro-2-methylphenoxyacetic acid (MCPA) from water by MIL-101(Cr) metal-organic framework: kinetics, isotherms and statistical models

We are pleased to resubmit our manuscript to your reputable journal. All the comments from the reviewers have been addressed as follows:

Referee: 1

Comment 1: (WHO) and (USEPA) abbreviations were mentioned in page 3, line 30 however what they stand for is mentioned later in line 34, it should be mentioned earlier with the first use of the abbreviations

Response 1: The full meaning of WHO and USEPA have been corrected and mentioned earlier before the abbreviations.

Comment 2: 4-chloro-2-methylphenoxyacetic” Acid is missing in the first line of the abstract.

Response 2: The full name has been corrected in the abstract.

Comment 3: It would be nice to add more discussion on MIL-101(Cr) and its application on relevant work and compared it to old used methods in the introduction.

Response 3: More discussion on MIL-101(Cr) have been added.

Comment 4: Page 4, line 42 (environmental benign should be environmentally).

Response 4: The correction has been effected.

Comment 5: Page 4, line 46 (hash should be harsh).

Response 5: The correction has been made.

Comment 6: Page 4, line 45 (expected to possessed should be possess)

Response 6: The correction has been made.

Comment 7: Page 3, line 46 punctuation mistake add a comma after Thus.

Response 7: The correction has been made.

Comment 8: 4- Page 5, line 31 (MLP-FF-ANN consist should be consists).

Response 8: The correction has been made.

Comment 9: Page 6, line 2 MIL-10(Cr) should be MIL-101(Cr).

Response 9: The correction has been made.

Comment 10: Page 7, line 8 (formed crystallite should be crystalline).

Response 10: The correction has been made.

Comment 11: Page 7, line 15 (there stages should be three).

Response 11: The correction has been made.

Comment 12: Page 9, line 30 (result show should be result shows).

Response 12: The correction has been made.

Comment 13: Page 12, line 6 (place play should be plays).

Response 13: The correction has been made.

Comment 14: Page 12, line 8 (results is should be results are).

Response 14: The correction has been made.

Comment 15: Page 14, line 50 (stages that represents should be represent).

Response 15: The correction has been made.

Comment 16: Page 16, line 5 (which indicate better should be indicates).

Response 16: The correction has been made.

Comment 17: Page 16, line 7 (the MOF retain should be retains).

Response 17: The correction has been made.

Comment 18: Table 7, MIL-101Cr) should be MIL-101(Cr) line 50 and many more English need to be revised thoroughly.

Response 18: Table 7 has been corrected as table 8 with MIL-101(Cr). The spellings and grammar have been revised.

Comment 19: Elaborate more in why increasing the temperature increases adsorption efficiency?

Response 19: More explanation and justifications have been provided. The Thermodynamic studies have also been included.

Comment 20: The experimental reaction was conducted between 25-40 °C and TGA showed a weight loss around 5-200 °C, so leaching test to know Cr concentration is recommended since Cr(III) is considered toxic and could encounter oxidizing conditions and the point of this paper is to remove a toxic contaminant from water.

Response 20: The observation is highly noted. Unfortunately, some Covid-19 cases were discovered in my University on the 19th October 2020 and all teaching and laboratory activities have been closed down. Also, Malaysia is experiencing the third wave of Covid-19 and the government has official closed down all universities until December and may be extended until January 2021. The closure of all laboratories and physical teaching activities made it impossible to carry out the experiment in **comment 20**. However, we have noted the observation and will effect it in our future work, since this study forms part of a postgraduate research.

Referee: 2

Comment 1 (Abstract): The author should consider to put numerical value of ΔG_{ads} in the abstract.

Response 1: Thermodynamic studies have been introduced to the work. Parameters such as Gibbs free energy change (ΔG°), enthalpy change (ΔH°) and entropy change (ΔS°) were discussed.

Comment 2 (Introduction): Perhaps some insight the latest inventive step that has been taken in this work. The explanation on the idea of inventive step in this studies could be interesting if it is included in the introduction. Novelty need to be highlighted in the last paragraph.

Response 2: The inventive step and novelty of this study have been mentioned in the introduction.

Comment 3 (Experimental section): What is the purity of MCPA used in this study?

Response 3: The purity of MCPA is mentioned in the materials and methods section.

Comment 4: Please state the percent yield of the synthesized MILs MOF.

Response 4: The percentage yield of the MIL-101(Cr) has been added.

Comment 5: I guess it would be best if the author compare the adsorption performance of the new MILs MOF from this study with the previous literature (in terms of % MCPA removal).

Response 5: The adsorption performance in comparison with previous literature is highlighted in Table 8

Comment 6: Please calculate the ΔG_{ads} value and explain the mode of adsorption. Propose the adsorption mechanism as well.

Response 6: The Gibbs free energy change (ΔG°), enthalpy change (ΔH°) and entropy change (ΔS°) were discussed using the thermodynamic model. The mode of adsorption and proposed adsorption mechanism have been discussed.

We are looking forward for your consideration

Regards,

Dr Hamza Ahmad Isiyaka

Dr Nonni Soraya Sambudi

Department of Fundamental and Applied Sciences.

Department of Chemical Engineering.

Universiti Teknologi PETRONAS.

UNIVERSITI TEKNOLOGI PETRONAS

INSTITUTE OF TECHNOLOGY PETRONAS SDN. BHD.

(Company No: 352875U) Wholly-owned subsidiary of PETRONAS

Main Campus : 32610 Seri Iskandar, Perak Darul Ridzuan, Malaysia. Tel : 605-368 8000 Fax : 605-365 4075

Branch Office : Advanced Technology And Innovation Centre (ATIC), LS-1-2, Enterprise 4, Technology Park Malaysia (TPM),
Lebuhraya Puchong - Sg. Besi, Bukit Jalil, 57000 Kuala Lumpur, Malaysia. Tel : 603-8994 1192 Fax : 603-8994 1193

www.utp.edu.my